# Ras-Related Protein Rab-32 and Thrombospondin 1 Confer Resistance to the EGFR Tyrosine Kinase Inhibitor Osimertinib by Activating Focal Adhesion Kinase in Non-Small Cell Lung Cancer

**DOI:** 10.3390/cancers14143430

**Published:** 2022-07-14

**Authors:** Zeinab Kosibaty, Odd Terje Brustugun, Inger Johanne Zwicky Eide, Georgios Tsakonas, Oscar Grundberg, Luigi De Petris, Marc McGowan, Per Hydbring, Simon Ekman

**Affiliations:** 1Department of Oncology and Pathology, Karolinska Institutet, 17164 Stockholm, Sweden; zkosibaty@gmail.com (Z.K.); georgios.tsakonas@ki.se (G.T.); luigi.depetris@ki.se (L.D.P.); per.hydbring@ki.se (P.H.); 2Section of Oncology, Drammen Hospital, Vestre Viken Hospital Trust, 3004 Drammen, Norway; ot.brustugun@gmail.com (O.T.B.); i.j.z.eide@gmail.com (I.J.Z.E.); 3Institute of Clinical Medicine, Faculty of Medicine, University of Oslo, 0315 Oslo, Norway; 4Department of Cancer Genetics, Institute for Cancer Research, Norwegian Radium Hospital, Oslo University Hospital, 0424 Oslo, Norway; mcgowan790@gmail.com; 5Thoracic Oncology Center, Karolinska University Hospital, 17164 Stockholm, Sweden; oscar.grundberg@regionstockholm.se; 6Akademiska Straket 1, BioClinicum J6:20, 17164 Solna, Sweden

**Keywords:** non-small cell lung cancer, exosomal RNA, osimertinib, transcriptome, epidermal growth factor receptor, Ras-related protein Rab-32, thrombospondin 1

## Abstract

**Simple Summary:**

Osimertinib is a third-generation EGFR tyrosine kinase inhibitor and the standard of care therapy for non-small cell lung cancer patients harboring EGFR-activating mutations. However, even for patients treated with osimertinib, resistance inevitably occurs leading to disease progression. Here, we utilized two osimertinib-resistant cell lines and investigated their RNA profiles. We found that Ras-related protein Rab-32 (RAB32) and thrombospondin 1 (THBS1) were upregulated and associated with resistance in osimertinib-resistant cells as well as in liquid biopsies from patients with disease progression following osimertinib treatment. Moreover, we found RAB32 and THBS1 to be mechanistically linked to activation of the focal adhesion pathway where combination of osimertinib with a FAK inhibitor resulted in a synergistic suppression of viability of osimertinib-resistant cells. Our findings propose a potential therapeutic strategy for overcoming acquired resistance to osimertinib in non-small cell lung cancer.

**Abstract:**

Treatment with the tyrosine kinase inhibitor (TKI) osimertinib is the standard of care for non-small cell lung cancer (NSCLC) patients with activating mutations in the epidermal growth factor receptor (EGFR). Osimertinib is also used in T790M-positive NSCLC that may occur de novo or be acquired following first-line treatment with other EGFR TKIs (i.e., gefitinib, erlotinib, afatinib, or dacomitinib). However, patients treated with osimertinib have a high risk of developing resistance to the treatment. A substantial fraction of the mechanisms for resistance is unknown and may involve RNA and/or protein alterations. In this study, we investigated the full transcriptome of parental and osimertinib-resistant cell lines, revealing 131 differentially expressed genes. Knockdown screening of the genes upregulated in resistant cell lines uncovered eight genes to partly confer resistance to osimertinib. Among them, we detected the expression of Ras-related protein Rab-32 (RAB32) and thrombospondin 1 (THBS1) in plasmas sampled at baseline and at disease progression from EGFR-positive NSCLC patients treated with osimertinib. Both genes were upregulated in progression samples. Moreover, we found that knockdown of RAB32 and THBS1 reduced the expression of phosphorylated focal adhesion kinase (FAK). Combination of osimertinib with a FAK inhibitor resulted in synergistic toxicity in osimertinib-resistant cells, suggesting a potential therapeutic drug combination for overcoming resistance to osimertinib in NSCLC patients.

## 1. Introduction

Lung cancer has one of the highest mortality rates of solid cancers, accounting for nearly one-fifth of all cancer-related deaths globally. Lung cancer is divided into the two major histological subgroups consisting of small cell lung cancer (SCLC) and non-small cell lung cancer (NSCLC), with the latter representing the higher incidence of all lung cancer cases. Patients diagnosed with advanced NSCLC are systemically treated with chemotherapy, immunotherapy, and targeted therapy, depending on the genetic background of the disease [1].

Activating mutations in the kinase domain of the gene encoding the epidermal growth factor receptor (EGFR) resulting in a constitutively active receptor, promoting sustained tumor cell growth and metastasis, account for up to 15% of NSCLC cases of the adenocarcinoma type in Caucasians, while the prevalence may be up to four times higher in Asians [2]. NSCLC patients with mutant EGFR are eligible for targeted therapy, with tyrosine kinase inhibitors (TKI) targeting activated EGFR. There are multiple EGFR TKIs available including first-generation EGFR TKIs, erlotinib and gefitinib, as well as second-generation EGFR TKIs, afatinib and dacomitinib.

Erlotinib and gefitinib are ATP-competitive inhibitors that are reversible inhibitors. Afatinib and dacomitinib are both covalent inhibitors, irreversibly binding to the kinase domain of EGFR [3,4,5,6,7,8]. Although all these compounds display a favorable clinical outcome compared to chemotherapy, therapy resistance is to be expected. Acquired resistance to first/second-generation EGFR-TKIs is most commonly mediated by a secondary mutation in the *EGFR* gene, resulting in a T790M substitution that occurs in approximately 60% of NSCLC patients upon progression on first-line treatment [9].

Osimertinib is a third-generation EGFR TKI targeting NSCLC with most activating EGFR variants as well as the resistance mutation T790M, which render refractoriness to first- and second-generation EGFR TKIs. Due to the fact of its activity against the EGFR T790M variant, osimertinib was approved in 2017 for clinical use as a second-line therapy after the failure of first-line EGFR TKIs. Moreover, multiple clinical trials reported nearly doubled median progression-free survival times in NSCLC patients with activating mutations in EGFR receiving osimertinib compared to patients receiving erlotinib or gefitinib, resulting in the Food and Drug Administration’s (FDA) and European Medicine Agency’s (EMA) approval of osimertinib as a first-line treatment in 2018 [10,11,12,13,14,15].

However, even for patients treated with osimertinib, resistance inevitably occurs, leading to disease progression. Resistance mechanisms include additional mutations in the EGFR kinase domain, such as C797S, as well as genetic aberrations in *MET; HER2; BRAF; PIK3CA; KRAS; CCND1; CCND2; CCNE1; CDK4; CDK6.* Approximately half of all resistant cases are of unknown molecular origin and may involve driver alterations in the RNA and/or protein landscape [16,17,18].

Here, we took a systematic approach to investigate whether specific alterations in the transcriptome of osimertinib refractory NSCLC could be directly linked to osimertinib resistance and whether such alterations could be therapeutically targeted in vitro.

## 2. Material and Methods

### 2.1. Cell Culture

EGFR mutant parental (P) cell lines (i.e., NCI-H1975P and HCC827P) and osimertinib-resistant (OR) cell lines (i.e., NCI-H1975OR and HCC827OR) were cultured in RPMI-1640 medium, supplemented with 10% supplemented fetal bovine serum (FBS) at 37 °C, 5% CO_2_. Cells were passaged when reaching 80% confluency. Along with numerous other molecular alterations, NCI-H1975P harbors the activating L858R mutation and the T790M gatekeeper mutation in the EGFR kinase domain, and HCC827P harbors an activating deletion in exon 19 (E746_A750del) in the EGFR kinase domain [19].

### 2.2. Chemicals and Antibodies

Osimertinib was purchased from Selleckchem (S7297), and FAK inhibitor 14 and FAK autophosphorylation inhibitor from Abcam (ab144503). The following antibodies were obtained from Cell Signaling Technology: anti-FAK (#3285), anti-Phos-FAK (Tyr397) (#3283), anti-paxillin (#2542), anti-Phos-paxillin (Tyr118) (#69363), anti-thrombospondin-1 (D7E5F) (#37879), anti-AKT3 (#4059), anti-COL5A1(#37304), anti-E-cadherin (#3195), anti-PARP (#9542), anti-caspase-3 (#9662), anti-cleaved PARP (Asp214) (#5625), anti-N-cadherin (D4R1H) (#13116), anti-E-cadherin (24E10) (#3195), anti-vimentin (D21H3) (#5741), anti-rabbit IgG HRP-linked (#7074P2), and anti-mouse IgG HRP-linked (#7076P2).

The anti-CADM1 (PA3-16744) and anti-BICC (PA5-116342) were purchased from Thermo Fisher. The anti-IGFBP7 (ab74169) and anti-PTPRM (ab231607) were purchased from Abcam. The anti-β-actin (A2228) and anti-RAB32 (HPA025731) were purchased from Sigma-Aldrich.

### 2.3. Cell Viability Assay

NCI-H1975P, HCC827P, NCI-H1975OR, and HCC827OR cells were plated at a density of 10,000 cells/mL in 96-well plates in the presence of 1, 10, 100, or 1000 nM osimertinib or in DMSO (control). After 72 h post-plating, the cell viability was measured in a luminometer through a CellTiter-Glo cell viability assay according to the manufacturer’s instructions (Promega Cat. #G7571). The final DMSO concentration in osimertinib-treated and control cell cultures was 0.1%.

### 2.4. Wound Healing Assay

The migratory ability was measured using a wound healing assay. NCI-H1975P, HCC827P, NCI-H1975OR, and HCC827OR cells were seeded at a density of 2 × 10^5^ cells/mL in a 24-well plate. Cells were grown to near confluent monolayers in medium containing 10% FBS. Perpendicular wounds were scratched using a sterile 10 μL pipette tip. The cells were then washed twice with warm PBS, and the scratched areas were assessed using computer-assisted microscopy. The cells were incubated in RPMI-1640 media + supplements until the cells covered the wound. The migration areas were calculated and quantitated using ImageJ software.

### 2.5. Western Blot

NCI-H1975P, HCC827P, NCI-H1975OR, and HCC827OR cells were lysed in M-PER Mammalian Protein Extraction Reagent (Thermo Fisher Scientific, Waltham, MA, USA, #78501). The protein concentrations were assessed using the Pierce BCA Protein Assay kit (Thermo Fisher Scientific, #23225). Cell lysates were electrophoresed in Invitrogen NuPAGE precast gels (Thermo Fisher Scientific, #EA0378BOX) and then transferred to polyvinylidene difluoride membranes using the iBlot gel transfer system (BIO-RAD Laboratories, Hercules, CA, USA). After blocking using Intercept^®^ (PBS) Blocking Buffer (LI-COR Biosciences, Lincoln, NE, USA), membranes were probed with primary antibody overnight at 4 °C, washed with PBS containing 0.1% Tween-20, and then incubated with an appropriate secondary antibody. The protein bands were visualized by SuperSignal West Femto Maximum Sensitivity Substrate (Thermo Fisher Scientific), and images were captured using an iBright imaging systems (Thermo Fisher Scientific). The protein bands were quantified using ImageJ 1.53a, Rasband, W.S., National Institutes of Health, Bethesda, MD, USA, https://imagej.nih.gov/ij/ (accessed on 22 May 2022), 1997–2018.

### 2.6. Transcriptome Analysis

For transcriptome analysis, biological duplicates of NCI-H1975P, NCI-H1975OR, HCC827P, and HCC827OR cells were trypsinized and washed in PBS followed by total RNA extraction using the mirVana miRNA isolation kit with phenol (ThermoFisher Scientific, #AM1560) according to the manufacturer’s instructions. Extracted total RNA displayed RNA integrity numbers in the range of 9.4 to 10.0 for the four cell lines. A volume of 3 μL of eluted total RNA was pre-amplified for six cycles before being loaded onto Clariom D Pico Assay human transcriptome arrays (ThermoFisher Scientific, #902925). Expressed transcripts were normalized using the Signal Space Transformation (SST-RMA) normalization method. Hierarchical clustering analysis, volcano plot analysis, and differential gene expression analysis were performed using the Transcriptome Analysis Console (TAC) 4.0.2 software (ThermoFisher Scientific, Waltham, MA, USA). Differential gene expression was defined as >2-fold expression, *p* < 0.05, and a false discovery rate (FDR) < 0.05 in progression versus the baseline of disease samples.

### 2.7. Functional Enrichment Analysis and TCGA Co-Expression Analysis

Kyoto Encyclopedia of Genes and Genomes (KEGG) pathway analysis was performed for common genes using the Database for Annotation, Visualization, and Integrated Discovery (DAVID) (https://david.ncifcrf.gov, (accessed on 1 April 2021)) with *p* < 0.05 and more than 5 genes. The cBioportal was used to investigate the co-expression of RAB32 and THBS1 mRNA from 586 primary tissues of lung adenocarcinoma in the TCGA, Firehose Legacy study. There was no information available regarding EGFR-TKI treatment of these tumors. Spearman and Pearson tests were used to evaluate the correlation of gene expression.

### 2.8. siRNA Library Screen and Knockdown

NCI-H1975P, HCC827P, NCI-H1975OR, and HCC827OR cells were transfected with a customized siRNA library (Thermo Fisher Scientific). The siRNA library included siRNAs targeting 24 genes in a 96-well plate with three independent siRNAs for each gene. As negative and positive controls, silencer select negative control and silencer select GAPDH were employed (Appendix A).

NCI-H1975P, HCC827P, NCI-H1975OR, and HCC827OR cells were transfected in a 96-well plate at a density of 10,000 cells/mL by Lipofectamine RNAiMAX reverse-transfection (Thermo Fisher Scientific, #13778030). At 72 h post-transfection, cells were scored for viability through the CellTiter-Glo Cell Viability Assay (Promega). The effect on cell viability was normalized to the silencer Select Negative Control #1. Eight selected genes were transfected with specific siRNAs of each gene for 72 h, including siIGFBP7, siCADM1, siCOL5A1, siAKT3, siPTPRM, siBICC1, siRAB32, and siTHBS1 in HCC827OR and NCI-H1975OR cells.

The transfection was performed with Lipofectamine RNAiMAX (Thermo Fisher Scientific) in accordance with the manufacturer’s protocol. The specific siRNAs targeting eight gene candidates were purchased from Thermo Fisher Scientific (Appendix A).

### 2.9. RNA Isolation and Quantitative Real-Time PCR (RT-qPCR)

Total RNA was extracted from NCI-H1975OR and HCC827OR cells using a RNeasy Mini Plus Kit (Qiagen, Hilden, Germany, #74134) in accordance with the manufacturer’s instructions. Complementary DNA (cDNA) was synthesized from total RNA using a high-capacity cDNA Reverse Transcription Kit (Thermo Fisher Scientific Waltham, MA, USA). RT-PCR was performed with a High Capacity cDNA Reverse Transcription Kit (Applied Biosystem by Thermo Fisher Waltham, MA, USA) on a CFX96 Touch Real-Time PCR detection system (BIO-RAD, Hercules, CA, USA) in accordance with the manufacturer’s instructions. The GAPDH gene was used for normalization. The specific primer pairs used for RT-PCR were purchased from IDT.

### 2.10. Patient Cohort and Sample Preparation

A total of 17 patients were included in the study. All patients were enrolled in the multicenter phase II TREM study and diagnosed with EGFR T790M-mutant NSCLC with a treatment history involving disease progression on minimum one first- or second-generation EGFR TKI. All patients were treated with osimertinib. Blood samples were drawn at treatment start and at disease progression. Plasma was separated through centrifugal isolation, 2000× *g* for 15 min and aliquoted to fresh 1 mL tubes. Samples were stored at −80 °C. The regional ethical committee approved sampling for this study (Dnr. 2016/710-31/1).

### 2.11. Exosome RNA Extraction

Seventeen exosomal RNA sample pairs (i.e., baseline and progression) were isolated at the Karolinska Institute. A volume of 1 mL plasma/sample point was centrifuged at 16,000× *g* for 10 min followed by processing using the ExoRNeasy serum plasma midi kit (Qiagen, Hilden, Germany) in accordance with the manufacturer’s instructions, and the RNA was eluted in 14 μL Rnase-free water. RNA samples (i.e., baseline and progression) were subjected to the RT-PCR protocol described above.

### 2.12. Ethics Statement

The study received ethical approval from the institutional review board at Karolinska University Hospital (registration number: 2016/944-31/1) and the Oslo North Regional Ethics Board (2015/181). Additional approval by Stockholm Medical Biobank was received (Bbk-01605). Written consent was provided by all patients. The study was conducted in accordance with the Declaration of Helsinki and the ICH Guideline for Good Clinical Practice and according to regulatory requirements.

### 2.13. Statistical Analysis

In this study, we used unpaired Student’s *t*-tests to compare the significant difference in two groups including cell viability, wound healing, quantitative Western blots, and relative mRNA expression levels. GraphPad Prism 9 software was used to carry out statistical analyses of the seventeen RNA sample pairs (i.e., baseline and progression), and paired Student’s *t*-test to calculate the significant differences between groups. The data are expressed as the mean ± SEM and statistical significance as *p*-values: * *p* < 0.05, ** *p* < 0.01, and *** *p* < 0.001.

## 3. Results

### 3.1. Characterization of Parental and Osimertinib-Resistant NSCLC Cell Lines

We utilized two EGFR-mutation-positive NSCLC cell lines, NCI-H1975 and HCC827, previously generated for refractoriness to osimertinib [20]. First, we examined their mutational status in the EGFR kinase domain through DNA Sanger sequencing, which displayed an identical genetic background between parental and resistant cell line pairs (data not shown).

We further determined the sensitivity of the cell lines to osimertinib. The viability of both parental cells was significantly diminished by >10 nM osimertinib for 72 h. In contrast, osimertinib-resistant cells were unaffected by 10 nM osimertinib treatment and largely tolerated up to 1000 nM osimertinib for 72 h (Figure 1A,B).

To determine the potential impact of osimertinib on apoptosis, we treated HCC827P, NCI-H1975P, HCC827OR, and NCI-H1975OR cells with 1000 nM osimertinib for 24 h followed by Western blot analysis of cleaved caspase-3 and PARP. We found that osimertinib effectively increased the levels of cleaved caspase-3 and PARP in parental cells, whereas minor changes were observed in resistant cells (Figure 1C).

### 3.2. Osimertinib-Resistant Cells Acquired an Enhanced EMT Phenotype

To explore the phenotypic changes in osimertinib-resistant cells, we compared the expression of epithelial-to-mesenchymal transition (EMT) markers in osimertinib-resistant cells lines to their parental counterparts. Resistant cells displayed a unique spindle cell-like morphology that was not observed in the parental cells (Figure 2A). Western blot analysis showed that the expression of E-cadherin was significantly decreased in resistant cells, whereas the expression of vimentin and N-cadherin increased in resistant cells (Figure 2B,C). Moreover, we examined cell motility in parental and osimertinib-resistant cells using a wound healing assay (Figure 2D) and found that resistant cells exhibited a faster wound healing capacity (Figure 2E), suggesting the acquisition of an EMT phenotype in osimertinib-resistant cells.

### 3.3. Transcriptome Profiling of Osimertinib-Resistant NSCLC Cells

We next sought to investigate the overall impact on the transcriptome landscape caused by osimertinib in NSCLC cells. We subjected our cell line panel to whole transcriptome analysis. A total of 1737 and 616 genes were differentially expressed in osimertinib-resistant cells compared to their parental counterparts in HCC827OR/P and NCI-H1975OR/P, respectively (Figure 3A,B and Appendix A).

When analyzing the differentially expressed genes through principal component analysis (PCA), we uncovered cell-line dependent variance structures where resistant cells clustered in a distinct fashion (Figure 3C). Venn diagram analysis revealed 131 genes to be differentially expressed in both cell line pairs, of which 26 genes were upregulated, 54 downregulated, and 51 genes were expressed in opposite directions in two cell line pairs in the resistant setting (Figure 3D and Appendix A).

### 3.4. siRNA Library Screening Analyses of Upregulated Genes in Osimertinib-Resistant Cells

In order to investigate whether differential expression would be indicative of a functional role in conferring resistance to osimertinib, we investigated all the upregulated genes through siRNA knockdown library screening, using three individual siRNAs of each gene, followed by cell viability scoring (Figure 4, Appendix A). We found that individual knockdown of eight different genes resulted in a >25% reduction of cell viability in osimertinib-resistant cells, insulin-like growth factor-binding protein 7 (IGFBP7), cell adhesion molecule 1 (CADM1), collagen-type V alpha 1 (COL5A1), v-akt murine thymoma viral oncogene homolog 3 (AKT3), protein tyrosine phosphatase receptor type M (PTPRM), BicC family RNA-binding protein 1 (BICC1), Ras-related protein Rab-32 (RAB32), and thrombospondin 1 (THBS1) (Table 1) and confirmed their knockdown at the mRNA and protein levels (Figure 5A,B).

### 3.5. RAB32 and THBS1 Were Elevated in Expression in NSCLC Patients with Disease Progression on Osimertinib

To validate the clinical relevance of our finding, we analyzed exosomal RNA from plasmas sampled at baseline, and again at progression of disease from 17 EGFR-mutant NSCLC patients enrolled in a multicenter phase II study (Appendix A).

Among the eight genes impacting viability in osimertinib-resistant cells, we detected mRNA expression of RAB32 and THBS1. The RT-qPCR results showed that the expression of both genes were significantly upregulated in progression samples compared to the baseline samples (Figure 6A). We further examined co-expression of RAB32 and THBS1 in lung adenocarcinoma cases from the cancer genome atlas (TCGA, Firehose legacy, 586 samples) and found that mRNA levels of RAB32 and THBS1 positively correlated with each other (Figure 6B).

### 3.6. RAB32 and THBS1 Are Mechanistically Linked to Focal Adhesion Kinase

When subjecting all the differentially expressed genes from our transcriptome profiling for KEGG-term analysis, we uncovered multiple pathways associated with focal adhesion (Figure 7A). We therefore hypothesized that RAB32 and THBS1 could confer resistance to osimertinib through regulation of focal adhesion signaling.

First, we confirmed RAB32 and THBS1 to be increased in expression in osimertinib-resistant cells (Figure 7B). Moreover, we found that the expression of phosphorylated focal adhesion kinase (FAK) (Tyr397) and phosphorylated paxillin (Tyr118) were elevated in osimertinib-resistant cells (Figure 7C), and that RAB32 and THBS1 knockdown diminished such phosphorylation events (Figure 7D). While osimertinib-resistant cells were completely refractory to single FAK inhibition or single osimertinib treatment, combination of the two drugs resulted in a significant reduction of cell viability in both of the resistant lines (Figure 7E,F).

## 4. Discussion

By combining two systematic approaches, whole genome transcriptomics and siRNA knockdown library screening, we identified multiple transcripts involved in osimertinib resistance in vitro. Two transcripts encoding for the small GTPase RAB32 and the glycoprotein thrombospondin-1 (THBS1) were further confirmed to be detected and upregulated in liquid biopsies sampled at disease progression from EGFR-mutation-positive NSCLC patients treated with osimertinib compared with liquid biopsies sampled at treatment baseline.

To our knowledge, this is the first report demonstrating a role of RAB32 and THBS1 in acquired resistance to osimertinib. Mechanistically, we linked RAB32 and THBS1 to the focal adhesion kinase (FAK) pathway and demonstrated that knockdown of either of these transcripts blunts activation of FAK, implicating that their upregulation in osimertinib refractory cells is necessary for FAK activation.

FAK, a nonreceptor tyrosine kinase, acts downstream of EGFR and has been demonstrated to be activated in ligand-dependent EGFR signaling [21,22,23] and to sustain MAPK and AKT signaling in conditions of EGFR inhibition in vitro [24].

Expression of THBS1 leads to increased binding to the calreticulin low-density lipoprotein receptor-related protein receptor complex, activation of ERK and PI3K, and the disassembly of focal adhesions, a critical step to enable cell motility and migration [25,26,27,28,29,30,31]. This biological outcome is blunted in FAK knockout fibroblasts [32], indicating a direct mechanistic link between THBS1 and FAK. THBS1 has been suggested to increase FAK activation, possibly by inactivation of the small GTPase RhoA [32]. The mechanistic connection between RAB32 and FAK remains elusive. RAB32 has been suggested to localize to the ER and mitochondria [33] and to regulate apoptosis and autophagy [34,35]. In addition, RAB32 has been reported to regulate cell size and proliferation, partly through interaction with the mTORC1 complex [36]. Since mTORC1 is downstream of FAK, there are likely additional functions of RAB32 in modulating FAK activity in osimertinib refractory cells.

We observed that knockdown of RAB32 or THBS1 was detrimental to osimertinib refractory cells, even in the absence of osimertinib, while sparing parental cells. In contrast, FAK inhibition had no impact on osimertinib refractory cells unless combined with osimertinib. This suggests that expression of RAB32 and THBS1 impacts multiple molecular processes, including FAK activation, of benefit for viability of osimertinib refractory cells, while increased FAK activity may solely compensate for blocked EGFR activity. Moreover, it indicates that the EGFR receptor is active in the absence of osimertinib and mechanistically accessible to osimertinib in our cell line systems. Hence, FAK and EGFR presents a synthetic lethal relationship, where cells refractory to osimertinib can cope with either one of them being shut down but not both. This highlights a potential to treat patients with disease progression during osimertinib therapy (Appendix A). The idea of combining EGFR inhibition with FAK inhibition for therapeutic purposes in lung cancer has been tested previously. Howe et al. demonstrated that combining erlotinib with different FAK inhibitors impacted cell viability in vitro and reduced tumor growth in vivo to a higher degree than single-agent treatment, correlating with reduced Akt phosphorylation [37]. Moreover, Solanki et al. demonstrated that treatment of smoke-exposed and TKI-resistant NSCLC cells with a FAK inhibitor restored their sensitivity to erlotinib, correlating with PI3K signaling activity [38]. It is possible that PI3K signaling also plays a mechanistic role in the signaling interplay between EGFR and FAK in osimertinib-resistant NSCLC. In addition to NSCLC and EGFR TKIs, FAK inhibitors have also been shown to synergize with other therapeutics, including chemotherapeutic drugs in pancreatic ductal adenocarcinoma [39] and breast cancer [40], and RAF/MEK inhibitors in multiple RAS-driven solid cancers [41].

The FAK inhibitor defactinib recently received an FDA breakthrough therapy designation for treatment of recurrent ovarian cancer in combination with RAF/MEK inhibition [42]. Moreover, there are numerous clinical trials recruiting cancer patients for defactinib treatment including multiple studies of NSCLC, emphasizing the potential of translating this finding to the clinic [43].

We also observed a diagnostic potential of RAB32 and THBS1 by profiling their RNA expression from exosomes extracted from longitudinally sampled plasma of EGFR-mutation-positive NSCLC patients treated with osimertinib. While this finding warrants further attention on RAB32 and THBS1 as potential markers of osimertinib resistance, our results need further validation in independent and larger clinical cohorts. It should also be mentioned that our liquid biopsy profiling cannot distinguish between tumor-derived exosomal RNA and exosomal RNA shed from healthy cells. While the field of liquid biopsies is increasingly developing with new methods, partly addressing the capturing of tumor-specific exosomes [44,45,46], our results would also need to be validated in longitudinally sampled tissue biopsies. Furthermore, a next step would be to validate the therapeutic potential of combining osimertinib and defactinib in mouse models, including state-of-the-art patient-derived xenograft (PDX) models of osimertinib refractory NSCLC.

## 5. Conclusions

In conclusion, we demonstrated that two transcripts, RAB32 and THBS1, conferred resistance to osimertinib by activating FAK signaling in NSCLC in vitro and possess potential as diagnostic biomarkers of osimertinib resistance in patients.

## Figures and Tables

**Figure 1 cancers-14-03430-f001:**
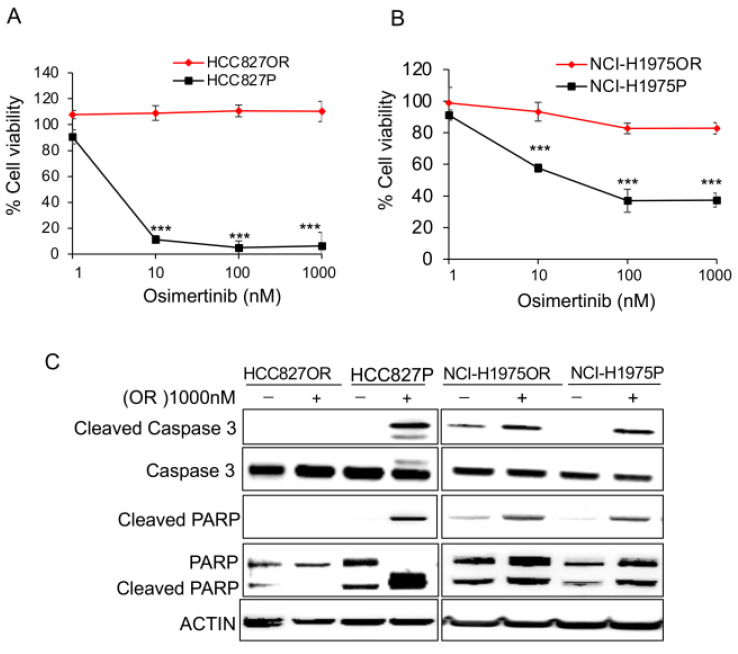
HCC827OR and NCI-H1975OR cells were resistant to osimertinib: (**A**) cell viability curve analysis of HCC827P and HCC827OR in increasing concentrations of osimertinib; (**B**) cell viability curve analysis of NCI-H1975P and NCI-H1975OR in increasing concentrations of osimertinib, where the *y*-axis displays the percentage of cell viability in comparison to DMSO; (**C**) HCC827OR, HCC827P, NCI-H1975OR, and NCI-H1975P were cultured and exposed to DMSO or 1000 nM osimertinib (OR) for 24 h, the cells were harvested for detection of apoptosis using Western blot with anti-cleaved caspase-3, anti-caspase-3, anti-cleaved PARP, and anti-PARP antibodies. β-Actin was used as the loading control. Full Western blot images can be found at Appendix A. Statistical significance was calculated through an unpaired two-tailed *t*-test. *** *p* < 0.001.

**Figure 2 cancers-14-03430-f002:**
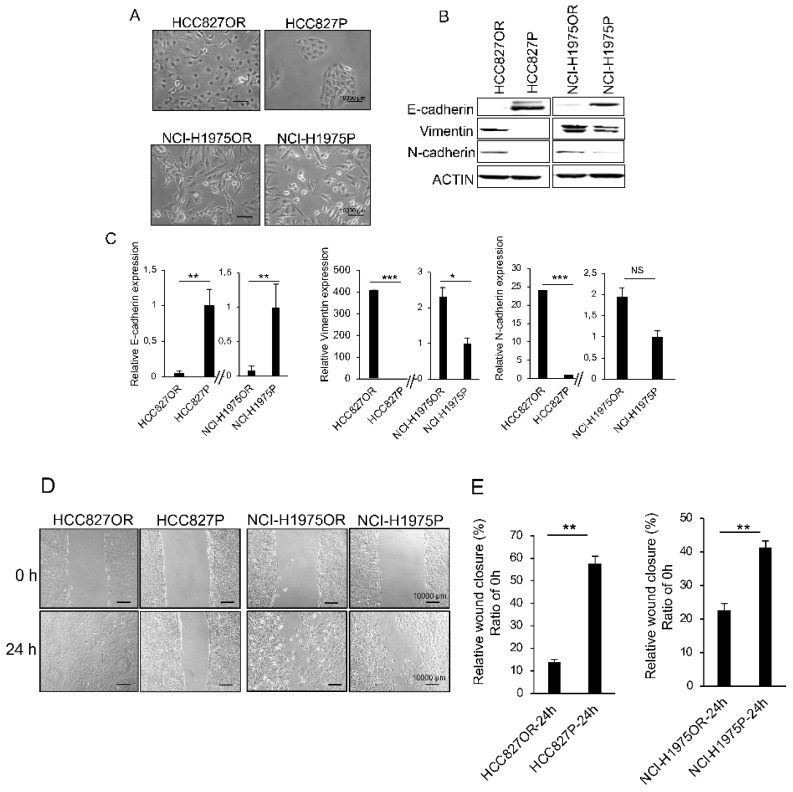
Acquisition of EMT phenotypes with enhanced cell motility in osimertinib-resistant cells. (**A**) Phase-contrast micrographs show parental cells (i.e., HCC827P and NCI-H1975P) and osimertinib-resistant cells (i.e., HCC827OR and NCI-H1975OR). Parental cells showed an epithelial appearance, whereas osimertinib-resistant cells displayed an elongated morphology. (**B**) Western blot analysis of EMT markers (i.e., E-cadherin, vimentin, and N-cadherin) in parental and osimertinib-resistant cells. (**C**) The band density of Western blot analysis from EMT markers was measured using ImageJ and normalized to β-actin. Error bars represent the mean ± SD from three independent experiments. (**D**) A wound healing assay was performed to measure cell motility in parental and osimertinib-resistant cells. (**E**) Relative wound closure was assessed and quantified using ImageJ software after 24 h of seeding the cells. Error bars represent the mean ± SD; six random microscopic fields were counted for each group from six independent experiments. Statistical significance was calculated through an unpaired two-tailed *t*-test. * *p* < 0.05, ** *p* < 0.01, and *** *p* < 0.001. NS: Not significant.

**Figure 3 cancers-14-03430-f003:**
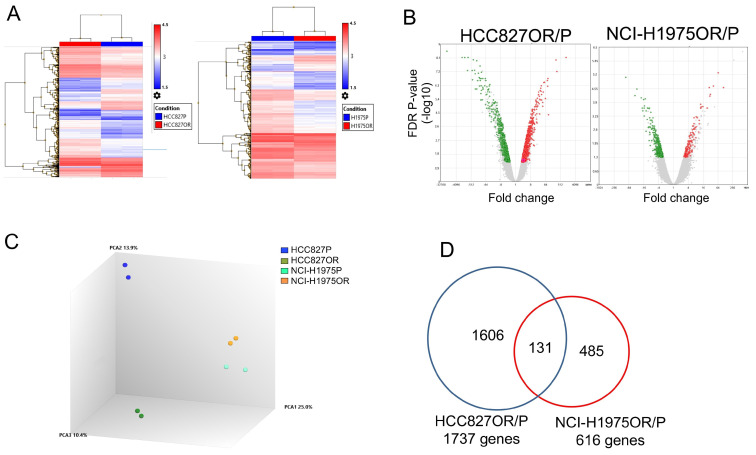
Mapping of the transcriptome in osimertinib-resistant cells. (**A**) Clustered heatmap showing a total of 1737 and 616 differentially expressed genes in osimertinib-resistant cell versus parental cells (i.e., HCC827OR/P and NCI-H1975OR/P, respectively). Each cell line was analyzed in biological duplicates. RNA expression is depicted on a scale from blue to red. (**B**) Volcano plot displaying downregulated transcripts (green) versus upregulated transcripts (red). (**C**) Three-dimensional principal component analysis (PCA) of the differentially expressed genes in HCC827P, NCI-H1975P, HCC827OR, and NCI-H1975OR. (**D**) Venn diagram showing 131 overlapping genes in HCC827OR/P and NCI-H1975OR/P.

**Figure 4 cancers-14-03430-f004:**
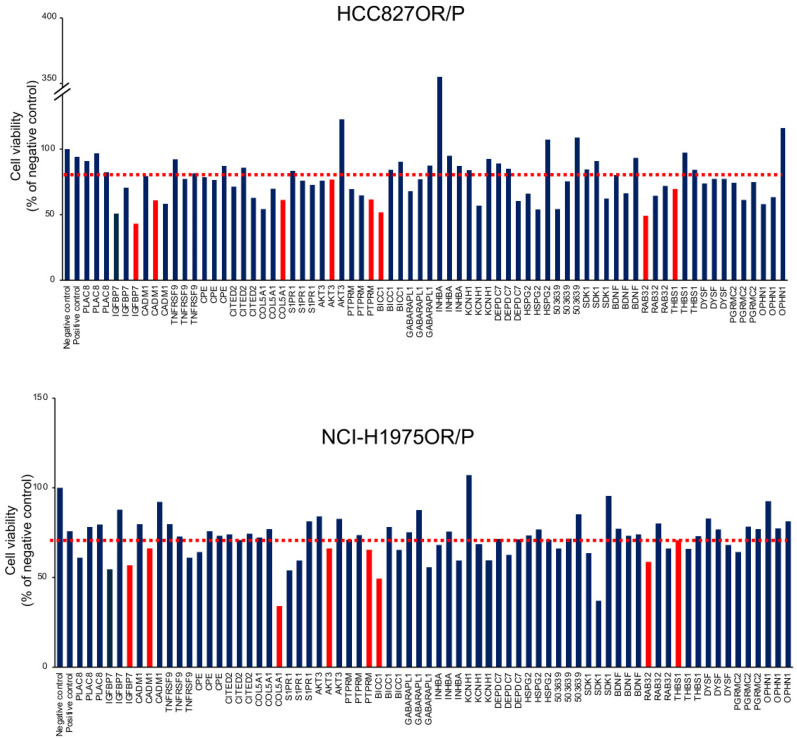
siRNA-library screening in HCC827OR versus HCC827 parental cells (HCC827OR/P) and NCI-H1975OR versus NCI-H1975 parental cells (NCI-H1975OR/P). Knockdown of 24 upregulated genes in HCC827OR/P (**upper** panel) and NCI-H1975OR/P (**lower** panel) using 3 sets of siRNAs for 72 h. Each bar visualizes cell viability for a given siRNA in comparison to cells transfected with a scrambled negative control. siRNAs specific for the GAPDH gene were employed as positive controls. Error bars represent the mean ± SD from three independent experiments. The cut-off of ∼25% reduction of cell viability is indicated by a red dotted line. The eight red bars represent the selected gene candidates with ∼25% reduction of cell viability in both HCC827OR/P and NCI-H1975OR/P cell line pairs. The blue bars represent 16 out of 24 genes tested with less than 25% reduction of cell viability in both cell line pairs.

**Figure 5 cancers-14-03430-f005:**
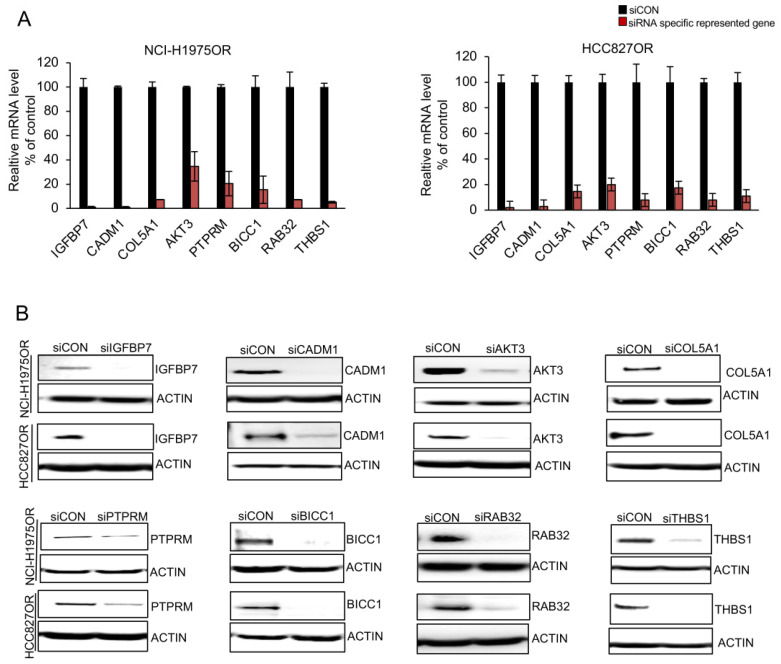
Suppression of specific siRNAs at mRNA and protein levels. Individual siRNAs targeting eight different genes (i.e., IGFBP7, CADM1, COL5A1, AKT3, PTPRM, BICC1, RAB32, and THBS1) were transfected in HCC827OR and NCI-H1975OR cells for 72 h. Knockdown efficiency was confirmed on an mRNA level using RT-qPCR (**A**) and protein level using Western blotting (**B**).

**Figure 6 cancers-14-03430-f006:**
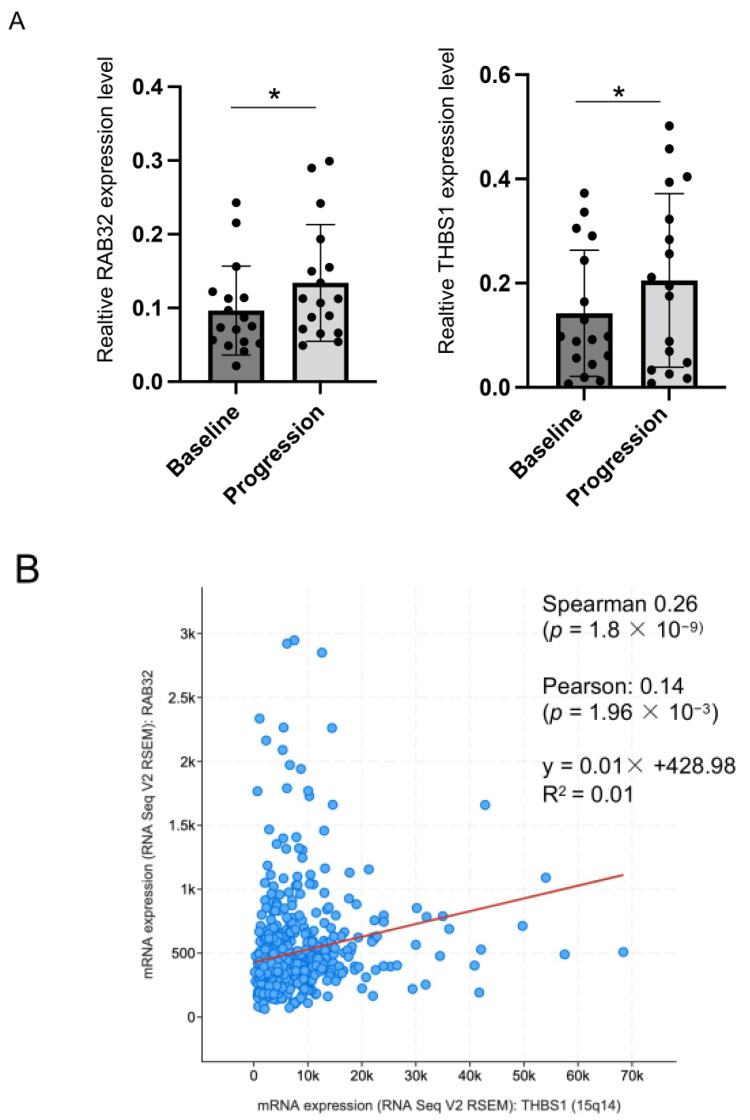
Elevated expression of RAB32 and THBS1 in plasma sampled from patients with disease progression on osimertinib. (**A**) Exosomal RNA extracted from the plasma of seventeen NSCLC patients, sampled at baseline versus progression of disease. Gene expression of RAB32 and THBS1 was assessed by RT-qPCR in 17 sample pairs, baseline and progression. Error bars represent the mean ± SD, n = 17. Statistical significance was calculated through a paired two-tailed *t*-test. * *p* < 0.05. (**B**) The association between RAB32 and THBS1 mRNA expression levels using a publicly available lung adenocarcinoma study (TCGA, Firehose legacy, 586 samples). Significant correlation was observed between the mRNA expression levels of RAB32 and THBS1.

**Figure 7 cancers-14-03430-f007:**
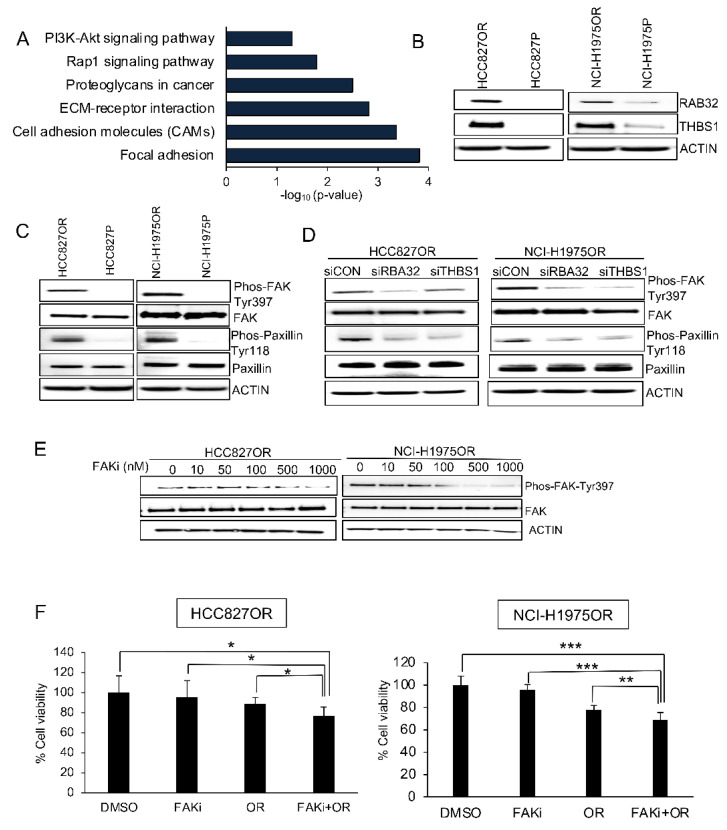
RAB32 and THBS1 were overexpressed and associated with focal adhesion signaling in osimertinib-resistant cell lines. (**A**) KEGG-term analysis of 131 overlapping genes in HCC827OR/P and NCI-H1975OR/P. Significant enrichment of KEGG pathways was defined as *p* < 0.05 and more than five genes. (**B**) Expression of RAB32 and THBS1 protein levels was examined by Western blotting in parental cells (i.e., HCC827P and NCI-H1975P) and osimertinib-resistant cells (i.e., HCC827OR and NCI-H1975OR). (**C**) Western blot analysis with Phos-FAK-Tyr397, FAK, Phos-Paxillin-Tyr118, and paxillin antibodies was performed on protein lysates from HCC827P, NCI-H1975P, HCC827OR, and NCI-H1975OR cells. (**D**) HCC827OR and NCI-H1975OR cells were transfected with siRAB32, siTHBS1, or siCON for 72 h and subjected to Western blotting for Phos-FAK-Tyr397, FAK, Phos-Paxillin-Tyr118, and paxillin. β-Actin was used as the loading control. (**E**) Serial concentration of focal adhesion kinase inhibitor (FAKi) was used to examine the inhibition of Phos-FAK-Tyr397 in HCC827OR and NCI-H1975OR cells. (**F**) Cell viability in HCC827OR and NCI-H1975OR after DMSO or (FAKi) (500 nM) and/or osimertinib (OR) (1000 nM) treatment. Error bars represent the mean ± SD, n = 5 for HCC827OR and n = 6 for NCI-H1975OR. Statistical significance was calculated through an unpaired two-tailed *t*-test. * *p* < 0.05, ** *p* < 0.01, and *** *p* < 0.001.

**Table 1 cancers-14-03430-t001:** Eight gene candidates based on siRNA library screening.

		HCC827OR/P				NCI-H1975OR/P			
Gene Symbol	Description	*p*-Value	FDR	Fold Change	Cell Viability% in HCC827OR/P	*p*-Value	FDR	Fold Change	Cell Viability % in NCI-H1975OR/P
IGFBP7	Insulin-like growth factor-binding protein 7	3.17 × 10^−11^	0	88.52	43.12%	1.31 × 10^−9^	2.55 × 10^−5^	28.58	56.81%
CADM1	Cell adhesion molecule 1	0.0000704	0.0075	5.15	60.90%	4.63 × 10^−8^	0.0002	24.01	66.24%
COL5A1	Collagen, type V, alpha 1	4.85 × 10^−9^	0	17.97	61.22%	8.95 × 10^−7^	0.0017	6.5	34.08%
AKT3	v-akt Murine thymoma viral oncogene homolog 3	0.0000884	0.0087	5.59	76.78%	7.43 × 10^−5^	0.0225	5.77	66.20%
PTPRM	Protein tyrosine phosphatase, receptor type, M	0	0.00006	10.6	61.58%	5.03 × 10^−6^	0.0046	5.43	65.46%
BICC1	BicC family RNA-binding protein 1	0.0002	0.0129	4.89	51.75%	9.91 × 10^−5^	0.0259	5.3	49.36%
RAB32	RAB32, member RAS oncogene family	0.0001	0.0115	3.47	49.18%	0.0002	0.035	3.32	58.71%
THBS1	Thrombospondin 1	0	0.00001	10.24	69.59%	4.88 × 10^−5^	0.0177	3.18	70.81%

## Data Availability

The data generated in this study are available within the article and its Appendix A. Raw data for this study were generated at the Karolinska Institute BEA core facility and are available from the corresponding author upon request.

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
