# Peer review of "Ras-Related Protein Rab-32 and Thrombospondin 1 Confer Resistance to the EGFR Tyrosine Kinase Inhibitor Osimertinib by Activating Focal Adhesion Kinase in Non-Small Cell Lung Cancer"

_cancers, 2022, doi:10.3390/cancers14143430_

Round 1
Reviewer 1 Report
Abstract
1. „Treatment with the tyrosine kinase inhibitor (TKI) osimertinib is the standard of care for non-small cell lung cancer (NSCLC) patients with activating mutations in the epidermal growth factor receptor (EGFR)” – please, provide more precise information; osimertinib is also used in T790M-positive NSCLC that may de novo or acquired following first-line treatment with other EGFR tyrosine kinase inhibitors (gefitinib, erlotinib, and afatinib).
Introduction
2. “Such resistance is often accompanied by additional mutations in the EGFR kinase domain, most commonly the T790M mutation, demonstrating the need for novel T790M EGFR TKIs” – please, provide more precise information; acquired resistance to 1-/2-generation EGFR-TKIs is mediated by secondary T790M mutation in EGFR gene that occur in approx. 60% of NSCLC patients upon progression on first line treatment.
3. “Osimertinib is a third generation EGFR TKI targeting NSCLC with most activating EGFR variants, including T790M” – T790M is not considered as activating EGFR variant as it is related to primary or acquired resistance to 1-/2-generation EGFR-TKIs.
Material and Methods
4. “2.1. Cell culture” – please, provide characteristics of known molecular alterations in both NCI-H1975P and HCC827P cell lines, in particular information about mutations in EGFR gene and other genes engaged in response to EGFR-TKIs, such as MET, PIK3CA, PTEN etc. ; preferably a supplementary table could be added.
5. “2.3. Cell viability assay” – what was final DMSO concentration in osimertinib-treated and control cell cultures?
6. Please, describe first how RNA was extracted from cell cultures before the paragraph on transcriptome analysis. Were the cells subjected to Trypsinization before RNA extraction? What was the quality of extracted RNA, e.g. RIN value?
7. “2.6. Transcriptome analysis” – it is not clear whether the transcriptome analysis was performed using total RNA extracted from NCI-H1975P/OR and HCC827P/OR cells or from exosomes purified from cell culture medium? If only exosomal RNA was analyzed, explain why this source of material was chosen. If RNA exctrated from cells was used for transcriptome analysis – please, add a paragraph and describe the method.
8. “2.9. RNA isolation and quantitative real-time PCR (RT-qPCR)” - was the same cellular RNA eluate used for transcriptome analysis by microarray and RT-qPCR?
9. “2.11. Exosome RNA extraction - Seventeen exosomal RNA samples (baseline and progression)” – seventeen RNA sample pairs (baseline and progression) were used?
10. Please, add new paragraph describing the method of analysis of lung adenocarcinoma cases from the cancer genome atlas (TCGA, Firehose legacy, 586 samples). What specimen was analyzed (tumor tissue or plasma), what point of treatment (before treatment, at progression), what EGFR-TKI treatment was used in this cohort of patients?
11. Where there any statistical calculations performed in this study? How data was presented, e.g. mean+SD or median (min-max)? Please, add another paragraph in this section.
Results
12. “3.3. Transcriptome profiling of osimertinib-resistant NSCLC cells. We subjected our cell line panel to whole transcriptome analysis” – this information should be included in Material and Methods section.
13. Figure 3. – please, provide Volcano plot for H1975P/OR cells.
14. “3.5. RAB32 and THBS1 are elevated in expression in NSCLC(…).We further examined co-expression of RAB32 and THBS1 in lung adenocarcinoma cases from the cancer genome atlas (TCGA, Firehose legacy, 586 samples) and found that mRNA levels of RAB32 and THBS1 positively correlated with each other (Figure 6B)” – What specimen was analyzed (tumor tissue or plasma), what point of treatment (before treatment, at progression), what EGFR-TKI treatment was used in this cohort of patients?
15. What was EGFR expression level in osimertinib-treated and control cells?
16. How about data from the transcriptome analysis of eluted exosomal total RNA?
Discussion
17. Please, discuss the data from the previous studies evaluating the anti-tumor activity of Focal Adhesion Kinase in non-small cell lung cancer cells resistant to EGFR-TKIs, in particular PMID: 29556515, PMID: 26962872, and provide some examples from studies on other cancer types.
18. Please, prepare a scheme of EGFR signaling presenting the role of RAB32, THBS1 and focal adhesion kinase (FAK) and include it in the supplementary material.
Author Response
Reviewer 1:
Abstract
- „Treatment with the tyrosine kinase inhibitor (TKI) osimertinib is the standard of care for non-small cell lung cancer (NSCLC) patients with activating mutations in the epidermal growth factor receptor (EGFR)” – please, provide more precise information; osimertinib is also used in T790M-positive NSCLC that may de novo or acquired following first-line treatment with other EGFR tyrosine kinase inhibitors (gefitinib, erlotinib, and afatinib).
Reply: thank you for this suggestion. In the abstract, we have added a sentence to clarify: “Osimertinib is also used in T790M-positive NSCLC that may occur de novo or be acquired following first-line treatment with other EGFR TKIs (gefitinib, erlotinib, afatinib or dacomitinib).”
Introduction
- “Such resistance is often accompanied by additional mutations in the EGFR kinase domain, most commonly the T790M mutation, demonstrating the need for novel T790M EGFR TKIs” – please, provide more precise information; acquired resistance to 1-/2-generation EGFR-TKIs is mediated by secondary T790M mutation in EGFR gene that occur in approx. 60% of NSCLC patients upon progression on first line treatment.
Reply: thank you for the clarification. In the Introduction, we have changed the sentence according to the reviewer´s suggestion: “Acquired resistance to first/second generation EGFR-TKIs is most commonly mediated by a secondary mutation in the EGFR gene resulting in a T790M substitution that occurs in approximately 60% of NSCLC patients upon progression on first-line treatment”.
- “Osimertinib is a third generation EGFR TKI targeting NSCLC with most activating EGFR variants, including T790M” – T790M is not considered as activating EGFR variant as it is related to primary or acquired resistance to 1-/2-generation EGFR-TKIs.
Reply: we thank the reviewer for this clarification. We have changed the sentence to: “Osimertinib is a third generation EGFR TKI targeting NSCLC with most activating EGFR variants as well as the resistance mutation T790M, which render refractoriness to first and second generation EGFR TKIs.”
Material and Methods
- “2.1. Cell culture” – please, provide characteristics of known molecular alterations in both NCI-H1975P and HCC827P cell lines, in particular information about mutations in EGFR gene and other genes engaged in response to EGFR-TKIs, such as MET, PIK3CA, PTEN etc. ; preferably a supplementary table could be added.
Reply: In response to the reviewer, we provide two tables (NCI-H1975 and HCC827, see end of pdf-attachment) outlining all known mutations in parental NCI-H1975 and parental HCC827, and reference the latest publication of the Cosmic database (new reference 19). The tables are downloaded from Cosmic v. 96 (www.cancer.sanger.ac.uk/cosmic). Since this information is freely available to academic scientists, we do not see the benefit of adding it as additional supplementary tables to the manuscript, but have instead included a reference (new reference 19) for the readers. For EGFR specifically, NCI-H1975 harbors the L858R and T790M mutations. HCC827 harbors a deletion in exon 19. L858R and del-exon 19 are considered activating EGFR mutations while T790M is a gatekeeper mutation. This information has been added in section “2.1. Cell culture”.
- “2.3. Cell viability assay” – what was final DMSO concentration in osimertinib-treated and control cell cultures?
Reply: The final DMSO concentration in osimertinib-treated and control cell cultures was 0.1%. We have included this information in Material and Methods, section 2.3
- Please, describe first how RNA was extracted from cell cultures before the paragraph on transcriptome analysis. Were the cells subjected to Trypsinization before RNA extraction? What was the quality of extracted RNA, e.g. RIN value?
Reply: We have extended the information in Material and Methods regarding RNA extraction and RNA quality measurement (including RIN value range) before total RNA was loaded for Clariom D transcriptome analysis.
- “2.6. Transcriptome analysis” – it is not clear whether the transcriptome analysis was performed using total RNA extracted from NCI-H1975P/OR and HCC827P/OR cells or from exosomes purified from cell culture medium? If only exosomal RNA was analyzed, explain why this source of material was chosen.If RNA exctrated from cells was used for transcriptome analysis – please, add a paragraph and describe the method.
Reply: We apologize for the confusion, which is due to an error in the description in this section. The transcriptome analysis was performed using total RNA extracted from NCI-H1975P/OR and HCC827P/OR and not exosomal total RNA. We have added a paragraph in Material and Methods “2.6. Transcriptome analysis” to describe the method of RNA extraction.
- “2.9. RNA isolation and quantitative real-time PCR (RT-qPCR)”- was the same cellular RNA eluate used for transcriptome analysis by microarray and RT-qPCR?
Reply: The RNA used for transcriptome analysis and the RNA used for RT-qPCR is distinct from each other, including distinct isolation methods. The RNA used for transcriptome analysis was extracted with a method enabling long as well as short RNA species to maximize RNA coverage across various RNA species. Hence, transcriptome analysis and quantitative real-time PCR is described in separate sections in the Material and Methods section.
- “2.11. Exosome RNA extraction - Seventeen exosomal RNA samples (baseline and progression)” – seventeen RNA sample pairs (baseline and progression) were used?
Reply: We analyzed seventeen RNA sample pairs, baseline and progression. We have added “Pairs” to the Material and Methods, section 2.11
- Please, add new paragraph describing the method of analysis of lung adenocarcinoma cases from the cancer genome atlas (TCGA, Firehose legacy, 586 samples). What specimen was analyzed (tumor tissue or plasma), what point of treatment (before treatment, at progression), what EGFR-TKI treatment was used in this cohort of patients?
Reply: The cBioportal was used to investigate the co-expression of RAB32 and THBS1 mRNA from 586 primary tissues of lung adenocarcinoma in the TCGA, Firehose legacy study. However, the TCGA, Firehose legacy study did not provide information about EGFR-TKI treatment. Spearman and Pearson tests were used to evaluate the correlation of gene expression from 586 primary tissues of lung adenocarcinoma in the TCGA. We have added a description about this analysis in Material and Methods, section 2.7
- Where there any statistical calculations performed in this study? How data was presented, e.g. mean+SD or median (min-max)? Please, add another paragraph in this section.
Reply: In this study we used unpaired Student’s t-test to calculate the significant differences in two groups of cell viability, wound healing, quantitative Western blots, and relative mRNA expression. Graphpad Prism 9 software was used to carry out statistical analyses in the seventeen RNA sample pairs (baseline and progression), and paired Student’s t-test to calculate the significant difference of the two groups. The data is represented as mean ± SEM and statistical significance as p value * p <0.05, ** p <0.01, *** p <0.001. We have added a paragraph in Material and Methods, section “2.13 Statistical analysis”
Results
- “3.3. Transcriptome profiling of osimertinib-resistant NSCLC cells. We subjected our cell line panel to whole transcriptome analysis” – this information should be included in Material and Methods section.
Reply: This information has now been clarified in Material and Methods, section “2.6 Transcriptome analysis”.
- Figure 3. – please, provide Volcano plot for H1975P/OR cells.
Reply: We have confirmed that the right panel, Volcano plot of NCI-H1975OR/P, is present in Figure 3B. We thank the reviewer for pointing this out.
- “3.5. RAB32 and THBS1 are elevated in expression in NSCLC(…).We further examined co-expression of RAB32 and THBS1 in lung adenocarcinoma cases from the cancer genome atlas (TCGA, Firehose legacy, 586 samples) and found that mRNA levels of RAB32 and THBS1 positively correlated with each other (Figure 6B)” – What specimen was analyzed (tumor tissue or plasma), what point of treatment (before treatment, at progression), what EGFR-TKI treatment was used in this cohort of patients?
Reply: The answer to this question was provided in the reply to question/issue #10, above, and in Material and Methods.
- What was EGFR expression level in osimertinib-treated and control cells?
Reply: According to the suggestion from the reviewer, we examined the expression of EGFR using Western blotting from osimertinib-treated cells (1000nM osimertinib for 24 hours) and control cells. The results showed that the expression of EGFR was similar in both parental cells and osimertinib-resistant cells regardless of presence of osimertinib, see image in attached word-document.
- How about data from the transcriptome analysis of eluted exosomal total RNA?
Reply: The transcriptome analysis (Clariom D) was conducted on total RNA from cell lines, NCI-H1975P/OR and HCC827P/OR. For the eluted exosomal total RNA, we performed targeted qPCR on RAB32 and THBS1 but no systematic transcriptome analysis. We apologize for the confusion in section ”2.6 Transcriptome analysis”, which was due to an error in the description of this section.
Discussion
- Please, discuss the data from the previous studies evaluating the anti-tumor activity of Focal Adhesion Kinase in non-small cell lung cancer cells resistant to EGFR-TKIs, in particular PMID: 29556515, PMID: 26962872, and provide some examples from studies on other cancer types.
Reply: We thank the reviewer for pointing out these important studies. We have included both studies and discussed their findings in the discussion section. Moreover, we have included three additional studies on FAK inhibition in other cancer types.
- Please, prepare a scheme of EGFR signaling presenting the role of RAB32, THBS1 and focal adhesion kinase (FAK) and include it in the supplementary material.
Reply: Model for RAB32-THBS1-mediated EGFR and FAK signaling in osimertinib-resistance of NSCLC cells. Overexpression of THBS1 in osimertinib- resistance promotes FAK autophosphorylation at Y397 and phosphorylation of paxillin at Tyr118. The activation of FAK signaling enhances Rho small GTPase and downstream oncogenic functions. Additionally, RAB32, a small GTPase is potentially enhanced via activation of a FAK signaling complex. Moreover, overexpression of RAB32 in osimertinib-resistance likely provides a feedback signal to activate FAK signaling. It is established that EGFR can activate Src and vice versa that Src activates EGFR. EGFR and FAK singling could simultaneously enhance osimertinib resistance leading to promotion of cell growth and migration. We have added a model for RAB32-THBS1-mediated EGFR and FAK signaling in Supplementary Figure 4.

Reviewer 2 Report
Dear authors:
I appreciated the present work that explored the DE genes involved in the osimertinib resistance from cell line studies. Impressively, a couple of viability tests using siRNA transfections were applied to identify the actual influential genes. After being validated with the patient’s cohort, RAD32 and THBS1 were thought to be meaningful. An associated FAK pathway was also explored via KEGG analysis. The authors concluded that RAD32 and THBS1 were upregulated in Osimertinib-resistant lung cancer, linking to the activation of the FAK pathway. This regulation might be a crucial resistant mechanism other than the acquired off-target driver mutations such as MET or HER2 alterations. Moreover, the combination of Osimertinib and FAK inhibitors would reverse the resistance to Osimertinib. I think it will be helpful to the management of lung cancer. However, some points should be addressed before the article can be published.
Major issues:
1. This article addressed an important issue that some resistant mechanisms to osimertinib may not involve driver mutations such as acquired MET or HER2 but something related to RNA or protein levels. The authors displayed the meaningful and reasonable molecules and pathways that may promote the resistance. However, I wondered whether the resistance mechanism involving RAD32, THBS1 and FAK are independent of the acquired driver mutations or not. In other words, could this regulation be found in patients who are resistant to osimertinib because of acquired MET or HER2 alterations? Did the authors analyze the resistant mechanisms among the 17 patients? Maybe they actually have any acquired driver mutation?
Minor issues:
Abstract
1. Line 30
The authors stated that “However, patients treated with Osimertinib have a high likelihood of developing resistance.” Nonetheless, almost every patient develops resistance to Osimertinib sooner or later. I’ll suggest the authors modify the ambiguous phrase “high likelihood.”
2. Line 37
The authors stated that “Both genes were amplified in progression samples.” Did the authors mean for RAB32 and THBS1 gene amplification? I think there is no testing performed for DNA sequencing, so the amplification would not be known. I’ll suggest replacing the term “amplified” with “upregulated” or “over-expressed.”
Introduction
3. Line 56,57
The authors reported the estimated EGFR mutation rates in NSCLC are about 15% in Caucasians and maybe up to 50-60% in Asians, respectively. However, the rates reported above are often calculated in the subpopulation of “adenocarcinoma” but not all NSCLC patients. I’ll suggest the authors clarify the prevalence of EGFR mutations and offer a reliable reference.
4. Line 62, A typo: “reversable” -> “reversible”
5. Line 71, “a second linetherapy for failed first line TKIs” -> “ a second line therapy after the failure to first line EGFR TKIs”
6. Line 71, “Following clinical trials reporting nearly doubled median progression-free survival times in patients receiving osimertinib compared to patients receiving erlotinib or gefitinib osimertinib received food and drug administration (FDA) and European medicine agency (EMA) approval in 2018 as first-line treatment for NSCLC patients with activating mutations in EGFR (9-14).” The paragraph needs further English checking.
Material and Methods
7. Line 147, a typo “a lung adencocarcinoma study” -> “a lung adenocarcinoma study”
8. Line 178. The authors state that “total of 17 patients were included in the study. All patients were enrolled in the multicenter phase II TREM study and diagnosed with EGFR T790M-mutant NSCLC with a treatment history involving disease progression on minimum one first-generation EGFR TKI.” However, the TREM study also enrolled patients who failed to afatinib, a second-generation TKI. Did the authors mean that all the 17 patients were not exposed to afatinib?
RESULTS
9. Figure 1B, the labels of the X-axis are missing.
10. Figure 1, figure legend. The statistical significance is expressed by asterisk symbols. However, the exact value of the symbols should be addressed. For example, * p<0.05.
11. Figure 2D is missing from the manuscript PDF file.
12. Error bars of Figure 2E are missing.
13. 3.3 Transcriptome profiling of osimertinib-resistant NSCLC cells
Line 248. Based on supplementary table 2, some differentially expressed genes are multiplicated. (For example, DGKD is counted nine times.) The discrimination may originate from the number of different transcripts or genes. Would the authors re-calculate and revise the actual number of DE genes according to the official gene symbols?
14. Line 250, The selection criteria of differentially expressed genes are mentioned in the section of Material and Methods. The authors may omit the criteria in the present section.
15. Line 254, According to supplementary table 3, there are 26 DE genes upregulated and 54 down-regulated, respectively in both HCC827 and H1975 cell lines. The remaining genes (n=51) are not expressed in the same way in the two cell lines. I’ll suggest the authors modify the reporting method. How about depicting 2 Venn diagrams? One for those genes upregulated in the resistant setting (the intersected number would be 26), and the other for genes down-regulated in the resistant setting (the intersected part would be n=54).
16. Figure 3B, the right panel is missing on the PDF file.
17. Figure 3D, Suggest revising the numbers after excluding the duplicates of genes.
18. Figure 4. There are 9 red bars, not 8 as the authors mentioned in the figure legend. Would the authors revise the figure? (Note that the CADM1 is not red, and KCNH1 should be blue.)
19. Figure 4, There should be a red dotted line to demonstrate the 75% cut-off as the authors mentioned in the legend. However, the red dotted line is not illustrated in this figure.
Author Response
Reviewer 2
Comments and Suggestions for Authors
Dear authors:
I appreciated the present work that explored the DE genes involved in the osimertinib resistance from cell line studies. Impressively, a couple of viability tests using siRNA transfections were applied to identify the actual influential genes. After being validated with the patient’s cohort, RAD32 and THBS1 were thought to be meaningful. An associated FAK pathway was also explored via KEGG analysis. The authors concluded that RAD32 and THBS1 were upregulated in Osimertinib-resistant lung cancer, linking to the activation of the FAK pathway. This regulation might be a crucial resistant mechanism other than the acquired off-target driver mutations such as MET or HER2 alterations. Moreover, the combination of Osimertinib and FAK inhibitors would reverse the resistance to Osimertinib. I think it will be helpful to the management of lung cancer. However, some points should be addressed before the article can be published.
Major issues:
- This article addressed an important issue that some resistant mechanisms to osimertinib may not involve driver mutations such as acquired MET or HER2 but something related to RNA or protein levels. The authors displayed the meaningful and reasonable molecules and pathways that may promote the resistance. However, I wondered whether the resistance mechanism involving RAD32, THBS1 and FAK are independent of the acquired driver mutations or not. In other words, could this regulation be found in patients who are resistant to osimertinib because of acquired MET or HER2 alterations? Did the authors analyze the resistant mechanisms among the 17 patients? Maybe they actually have any acquired driver mutation?
Reply: We thank the reviewer for this interesting point. However, re-biopsy was not mandatory in the clinical study and hence tissue or cytology is not available for the patients in our study to be able to analyze any co-occurring acquired genetic aberrations.
Minor issues:
Abstract
- Line 30
The authors stated that “However, patients treated with Osimertinib have a high likelihood of developing resistance.” Nonetheless, almost every patient develops resistance to Osimertinib sooner or later. I’ll suggest the authors modify the ambiguous phrase “high likelihood.”
Reply: In response to the reviewer, we have modified the phrase from “high likelihood” to “high risk”.
- Line 37
The authors stated that “Both genes were amplified in progression samples.” Did the authors mean for RAB32 and THBS1 gene amplification? I think there is no testing performed for DNA sequencing, so the amplification would not be known. I’ll suggest replacing the term “amplified” with “upregulated” or “over-expressed.”
Reply: We thank the reviewer for picking this up. As the reviewer points out, there was no analysis of DNA copy number, and hence, the wording is incorrect. We have replaced the term “amplified” with “upregulated”.
Introduction
- Line 56,57
The authors reported the estimated EGFR mutation rates in NSCLC are about 15% in Caucasians and maybe up to 50-60% in Asians, respectively. However, the rates reported above are often calculated in the subpopulation of “adenocarcinoma” but not all NSCLC patients. I’ll suggest the authors clarify the prevalence of EGFR mutations and offer a reliable reference.
Reply: thank you for this clarification. We have modified the sentence accordingly: “…..account for up to 15% of NSCLC cases of adenocarcinoma type in Caucasians while the prevalence may be up to four times higher in Asians.” We also added a new reference for this statement (ref no. 2: Zhang Y., Yuan J., Wang K., Fu X., Han X., Threapleton D., Yang Z., Mao C., Tang J. The prevalence of EGFR mutation in patients with non-small cell lung cancer: a systematic review and meta-analysis. Oncotarget. 2016; 7: 78985-78993. https://www.oncotarget.com/article/12587/text/.
- Line 62, A typo: “reversable” -> “reversible”
Reply: This typo has been corrected.
- Line 71, “a second linetherapy for failed first line TKIs” -> “ a second line therapy after the failure to first line EGFR TKIs”
Reply: This sentence has been corrected according to the suggestion from the reviewer.
- Line 71, “Following clinical trials reporting nearly doubled median progression-free survival times in patients receiving osimertinib compared to patients receiving erlotinib or gefitinib osimertinib received food and drug administration (FDA) and European medicine agency (EMA) approval in 2018 as first-line treatment for NSCLC patients with activating mutations in EGFR (9-14).” The paragraph needs further English checking.
Reply: We have modified this paragraph.
Material and Methods
- Line 147, a typo “a lung adencocarcinoma study” -> “a lung adenocarcinoma study”
Reply: This typo has been corrected.
- Line 178. The authors state that “total of 17 patients were included in the study. All patients were enrolled in the multicenter phase II TREM study and diagnosed with EGFR T790M-mutant NSCLC with a treatment history involving disease progression on minimum one first-generation EGFR TKI.” However, the TREM study also enrolled patients who failed to afatinib, a second-generation TKI. Did the authors mean that all the 17 patients were not exposed to afatinib?
Reply: we thank the reviewer for correctly pointing this out. It is true that the TREM study, including the patient cohort in this manuscript, also included patients progressing on the second-generation TKI afatinib according to the inclusion criterion: “Radiological disease progression following at least one prior EGFR TKI.” We have modified the sentence accordingly: “….disease progression on minimum one first- or second-generation EGFR TKI.”
RESULTS
- Figure 1B, the labels of the X-axis are missing.
Reply: We have confirmed that the labels displayed on the X-axis in Figure 1B are present.
- Figure 1, figure legend. The statistical significance is expressed by asterisk symbols. However, the exact value of the symbols should be addressed. For example, * p<0.05.
Reply: We have added the exact value of the symbols in which the statistical significance is calculated through an unpaired two-tailed t test, *** p <0.001.
- Figure 2D is missing from the manuscript PDF file.
Reply: We have included figure 2D in Results, section 3.2
- Error bars of Figure 2E are missing.
Reply: We have included error bars in figure 2E.
- 3.3 Transcriptome profiling of osimertinib-resistant NSCLC cells
Line 248. Based on supplementary table 2, some differentially expressed genes are multiplicated. (For example, DGKD is counted nine times.) The discrimination may originate from the number of different transcripts or genes. Would the authors re-calculate and revise the actual number of DE genes according to the official gene symbols?
Reply: Thank you for the suggestion. We revised the actual number of DE according to the official gene symbols and excluded some differentially expressed genes that were multiplicated. We found that a total of 1737 and 616 genes were differentially expressed in osimertinib-resistant cells compared to their parental counterparts in HCC827OR/P and NCI-H1975OR/P, respectively. The data were described in Results, section 3.3 and in supplementary table 2.
- Line 250, The selection criteria of differentially expressed genes are mentioned in the section of Material and Methods. The authors may omit the criteria in the present section.
Reply: In response to the reviewer, we have omitted this information from the present section.
- Line 254, According to supplementary table 3, there are 26 DE genes upregulated and 54 down-regulated, respectively in both HCC827 and H1975 cell lines. The remaining genes (n=51) are not expressed in the same way in the two cell lines. I’ll suggest the authors modify the reporting method. How about depicting 2 Venn diagrams? One for those genes upregulated in the resistant setting (the intersected number would be 26), and the other for genes down-regulated in the resistant setting (the intersected part would be n=54).
Reply: We provided Venn diagrams showing 26 upregulated and 54 downregulated overlapping genes in HCC827OR/P and NCI-H1975OR/P in Supplementary Figure 3 and the description was added in Results, section 3.3
- Figure 3B, the right panel is missing on the PDF file.
Reply: We have confirmed that the right panel is present in Figure 3B.
- Figure 3D, Suggest revising the numbers after excluding the duplicates of genes.
Reply: We re-analyzed the differentially expressed genes after excluding the duplicates of genes. Venn diagram analysis revealed 131 genes to be differentially expressed in both cell line pairs., Figure 3D.
- Figure 4. There are 9 red bars, not 8 as the authors mentioned in the figure legend. Would the authors revise the figure? (Note that the CADM1 is not red, and KCNH1 should be blue.)
Reply: According to the reviewer suggestion, we revised figure 4 and included only 8 red bars.
- Figure 4, There should be a red dotted line to demonstrate the 75% cut-off as the authors mentioned in the legend. However, the red dotted line is not illustrated in this figure.
Reply: Thank you for bringing up this point. We have confirmed that the red dotted line is displayed in figure 4 as a 75% cut-off line.

Round 2
Reviewer 2 Report
The authors have answered my questions and revised accordingly. Thanks.